# Beyond Target Networks: Improving Deep $Q$-learning with Functional Regularization

## Abstract

Much of the recent successes in Deep Reinforcement Learning have been based on minimizing the squared Bellman error. However, training is often unstable due to fast-changing target $Q$-values, and target networks are employed to stabilize training by using an additional set of lagging parameters. Despite their advantages, target networks can inhibit the propagation of newly-encountered rewards which may ultimately slow down training. In this work, we address this issue by augmenting the squared Bellman error with a functional regularizer. Unlike target networks, the regularization we propose here is explicit and enables us to use up-to-date parameters as well as control the regularization. This leads to a faster yet more stable training method. Across a range of Atari environments, we demonstrate empirical improvements over target-network based methods in terms of both sample efficiency and performance. In summary, our approach provides a fast and stable alternative to replace the standard squared Bellman error.

## 1 Introduction

In practice, Deep $Q$-learning (DQL) methods demonstrate instability due to being trained with off-policy data, function-approximations, and bootstrapping (Sutton & Barto, 2018; Van Hasselt et al., 2018). The Bellman error, which is used to train DQL methods, is computed with target $Q$-values estimated by a Deep Neural Network (DNN) with *constantly* changing parameters which results in instability. Target networks were proposed as a solution to stabilize training by performing an implicit regularization. They do so by using an additional set of parameters which can be updated in two different ways. The first way, which is a popular method for visual tasks such as Atari, is to *periodically* update the parameters to estimate the target $Q$-values (Mnih et al., 2013). The second way, popular in robotics and control tasks, is to parametrize the target network using a *moving average* of the parameters (Lillicrap et al., 2015).

Stabilizing the squared Bellman error using target networks parametrized by an additional set of weights has become standard practice in DQL algorithms achieving state-of-the-art performance in a variety of difficult tasks (Mnih et al., 2013; Lillicrap et al., 2015; Abdolmaleki et al., 2018b; Haarnoja et al., 2018a; Fujimoto et al., 2018; Hausknecht & Stone, 2015; Van Hasselt et al., 2016). Despite the impressive performance of target networks, using additional sets of parameters to estimate the target $Q$-values can be problematic. First, due to lagging parameters, newly encountered rewards may not be quickly propagated through the state space, ultimately slowing down the training. Second, while this can be partially addressed by using a moving-average (Polyak averaging) of parameters, such averaging procedure can be seen as a type of weight-regularization which too can slow down the updates. The weight-regularization is known to be insufficient for DNNs and do not necessarily improves stability (Benjamin et al., 2018). The goal of this paper is to address these issues by using up-to-date $Q$-values in the square Bellman error and using explicit $Q$-values regularization by functional regularization.

We propose an alternative to the squared Bellman error based on functional regularization which, as we show, enables us to go beyond the traditional target networks. Our approach allows us to use the up-to-date parameters to compute the Bellman error while stabilizing the training by functionally regularizing against a periodically updated prior network (see Fig. 1). Using up-to-date parameters addresses the issue of slow propagation of newly encountered rewards. Since the regularization is explicit, it enables us to tune the regularization level for a given task, which is not possible with

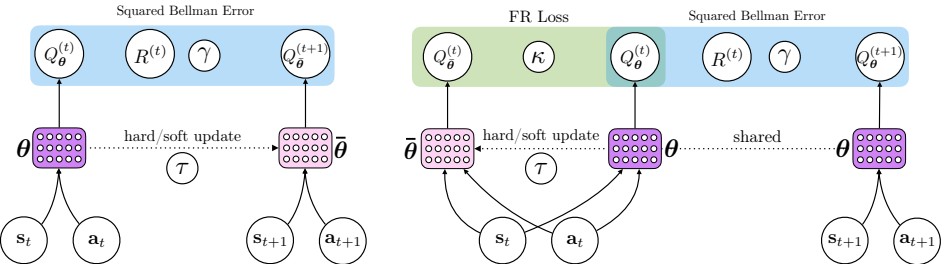

(a) The Bellman error with Target Networks.

(b) The Functionally Regularized Bellman error using our Prior Network approach.

Figure 1: (a) Target networks use an additional set of parameters, $\bar{\boldsymbol{\theta}}$ (pink), to estimate target values which are then used in the Bellman error. The target network stabilizes the $Q$-value estimates, but is not up-to-date. (b) In our approach, we use the up-to-date $\boldsymbol{\theta}$ to estimate the Q-value for the Bellman error (see the rightmost block), and add a $\kappa$ weighted Functional Regularization (FR) loss to stabilize the current $Q$-value estimate (parameterized by $\boldsymbol{\theta}$) by comparing it to the old $Q$-values (parameterized by $\bar{\boldsymbol{\theta}}$).

standard approaches that rely on implicit regularization based on target networks. We validate our new objective on the Four Rooms environment (Sutton et al., 1999) with DNNs function approximation and show that we can approximate the true value function and learn quickly. Finally, we demonstrate that DQL methods with the FR Bellman error outperform popular algorithms based on the Bellman error on a subset of the Atari suite (Bellemare et al., 2013) which is the most common benchmark for DQL methods.

## 2 BACKGROUND

**Preliminaries** We consider the general case of a Markov Decision Process (MDP) defined by $\{\mathcal{S}, \mathcal{A}, P, R, \gamma, \mu\}$, where $\mathcal{S}$ and $\mathcal{A}$ respectively denote the finite state and action spaces, $P \colon \mathcal{S} \times \mathcal{A} \times \mathcal{S} \to \mathbb{R}$ represent the environment transition dynamics, where $P(\cdot|\mathbf{s}, \mathbf{a})$ is the distribution of the next state taking action $\mathbf{a}$ in state $\mathbf{s}$. $R \colon \mathcal{S} \times \mathcal{A} \to \mathbb{R}$ denotes the reward function, $\gamma \in [0, 1)$ is the discount factor, and $\mu$ is the initial state distribution. We use an MDP to represent the sequential interaction between an agent and the environment, where at each round $t$ the agent observes its current state $\mathbf{s}_t$, selects an action $\mathbf{a}_t$ which results in a reward $R(\mathbf{s}_t, \mathbf{a}_t)$ and the next state $\mathbf{s}_{t+1} \sim P(\cdot|\mathbf{s}_t, \mathbf{a}_t)$. The reinforcement learning (RL) objective involves defining an agent that maximizes the expected discounted sum of rewards $\mathbb{E}_{P,\pi} \sum_t \gamma^t R(\mathbf{s}_t, \mathbf{a}_t)$, (with $\mathbf{s}_0 \sim \mu$), by means of a policy $\pi(\mathbf{a}|\mathbf{s})$, that given a state selects in expectation the best action. Bellman optimality equations are defined as:

$$Q^*(\mathbf{s}, \mathbf{a}) \equiv R(\mathbf{s}, \mathbf{a}) + \mathbb{E}_{P,\pi^*} \left[ \sum_{t=1}^{\infty} \gamma^t R(\mathbf{s}_t, \mathbf{a}_t) \middle| \mathbf{s}_0 = \mathbf{s}, \mathbf{a}_0 = \mathbf{a} \right], \quad (1)$$

$$= R(\mathbf{s}, \mathbf{a}) + \gamma \mathbb{E}_{\mathbf{s}', \mathbf{a}' \sim P, \pi^*} \left[ Q^*(\mathbf{s}', \mathbf{a}') \right], \quad (2)$$

where the optimal policy is defined as $\pi^*(\mathbf{a}|\mathbf{s}) = \delta(\mathbf{a} - \arg\max_{\mathbf{a}} Q^*(\mathbf{s}, \mathbf{a}))$, $\delta(\cdot)$ is the Dirac delta function, and we use $\pi^*$ to evaluate the expectations in Eqs. (1) and (2) (see (Watkins, 1989)). Note that the value function can be obtained as $V^*(\mathbf{s}_t) = \max_{\mathbf{a}} Q^*(\mathbf{s}_t, \mathbf{a})$.

$Q$-**value Estimation** We can estimate the $Q$-value for each state-action pair by formulating it as a regression problem, e.g., treating the right hand side of Eq. (1) as the target. However, we generally do not have access to the optimal policy or to samples from the optimal return. Fortunately, it is possible to learn from arbitrary policy by using the $Q$-learning algorithm (Watkins, 1989). Specifically, we can minimize the one-step Mean Squared Bellman Error for a given transition $(\mathbf{s}_t, \mathbf{a}, r_t, \mathbf{s}_{t+1})$ (Sutton, 1988):

$$l(Q) = \frac{1}{2} \bigg( \underbrace{R(\mathbf{s}_t, \mathbf{a}_t) + \gamma \mathbb{E}_{\mathbf{s}_{t+1}, \mathbf{a}_{t+1} \sim P, \pi} \left[ Q(\mathbf{s}_{t+1}, \mathbf{a}_{t+1}) \right]}_{\text{target } Q\text{-value}} - Q(\mathbf{s}_t, \mathbf{a}_t) \bigg)^2, \quad (3)$$

where $\pi(\mathbf{a}|\mathbf{s}) = \delta(\mathbf{a} - \arg\max_{\mathbf{a}} Q(\mathbf{s}, \mathbf{a}))$, and the loss is then averaged over multiple transitions. This approach of using future value estimates as regression targets is referred to as *bootstrapping*. Sometimes, the target is expressed using the Bellman operator $\mathcal{T}Q(\mathbf{s}_t, \mathbf{a}_t) = R(\mathbf{s}_t, \mathbf{a}_t) + \gamma\mathbb{E}_{\mathbf{s}_{t+1}, \mathbf{a}_{t+1} \sim P, \pi}[Q(\mathbf{s}_{t+1}, \mathbf{a}_{t+1})]$.

**Function Approximation**   The original Q-learning algorithm by Watkins (1989), estimates each state-action pair separately, is a prohibitive task in real-world applications. Instead, it is possible to approximate $Q$ by means of a parametrized function $Q_{\boldsymbol{\theta}}$ (e.g., a deep network), which can be learned by minimizing Eq. (3) with an iterative algorithm such as gradient descent. Unfortunately, combining off-policy data, function approximation, and bootstrapping makes learning unstable and potentially diverges (Van Hasselt et al., 2018; Sutton & Barto, 2018; Achiam et al., 2019). The problem arises when the parameters of the $Q$-network are updated to better approximate the $Q$-value of a state-action pair at the cost of worsening the approximation of other $Q$-values, including the ones used as targets.

One approach to stabilize learning involves estimating the regression targets with a *target network* $Q_{\bar{\boldsymbol{\theta}}}$, a copy of $Q_{\boldsymbol{\theta}}$ where $\bar{\boldsymbol{\theta}}$ is periodically updated only after a certain number of iterations (Mnih et al., 2013). This causes regression targets to no longer directly depend on the most recent estimate of $\boldsymbol{\theta}$. With this, the Mean Squared Projected Bellman Error (we will refer to this loss as simply the Squared Bellman Error as we only use functional approximation) loss for a given state transition is:

$$l^{\text{target}}(\boldsymbol{\theta}) = \frac{1}{2}\left(R(\mathbf{s}_t, \mathbf{a}_t) + \gamma\mathbb{E}_{\mathbf{s}_{t+1}, \mathbf{a}_{t+1} \sim P, \pi}\left[Q_{\bar{\boldsymbol{\theta}}}(\mathbf{s}_{t+1}, \mathbf{a}_{t+1})\right] - Q_{\boldsymbol{\theta}}(\mathbf{s}_t, \mathbf{a}_t)\right)^2. \tag{4}$$

A common strategy to stabilize learning is to update the target parameters only after a fixed number of training iterations i.e., periodic update (Mnih et al., 2013). In between these updates, the target is using the lagging parameters, and the newly encountered rewards are not immediately reflected in the estimated target $Q$-values. The delay in the reward propagation can, however, slow down training. For instance, consider a Markov chain environment of $N$ states with deterministic transitions, in which all states have reward of $0$ except the terminal, right-most state which has a reward of $1$. Consider using tabular $Q$-values and the corresponding target $Q$-values, which are updated every $H$ training steps. Even with access to all the state-action pairs, this scheme will require $NH$ training steps to propagate the right-most reward through the chain, i.e., to the state $N$ steps away. We can observe that the learning speed would be a direct function of $H$.

An alternative to hard-updating is the use of a moving average of the weights to parametrize the target $Q$-network as done with Polyak's averaging ("the soft update") (Lillicrap et al., 2015). In this case, the target network's weights are updated as $\bar{\boldsymbol{\theta}} \leftarrow (1 - \tau)\bar{\boldsymbol{\theta}} + \tau\boldsymbol{\theta}$, with $\tau \in (0, 1)$. However, $\boldsymbol{\theta}$ is not used to estimate the target $Q$-values, thus, our regression targets are no longer up-to-date and learning is slowed.

**Weight Regularization in DQN**   Soft updates can be seen as a simple *weight regularization* where the update of $\bar{\boldsymbol{\theta}} \leftarrow \boldsymbol{\theta}$ is delayed by adding an $L_2$ regularizer $\tau\|\boldsymbol{\theta} - \bar{\boldsymbol{\theta}}\|^2$. Essentially, we do not want the value of $\bar{\boldsymbol{\theta}}$ to abruptly change. This approach to improving stability is common in non-stationary time-series analysis (Brown, 1959; Holt et al., 1960; Gardner Jr, 1985), online learning (Cesa-Bianchi & Lugosi, 2006), and continual learning (Kirkpatrick et al., 2017; Nguyen et al., 2017). Recent attempts to go beyond target networks employ preconditioning methods which can also be seen as weight-regularization methods (Knight & Lerner, 2018; Achiam et al., 2019). For example, natural-gradient descent used in Knight & Lerner (2018) employs the Kullback-Leibler (KL) for Gaussian distributions $\mathcal{N}(q|Q_{\boldsymbol{\theta}}(s, a), 1)$ over the Q-values $q$, giving rise to a quadratic weight-regularizer. Another approach of Achiam et al. (2019) uses a different preconditioner which can also be seen as a quadratic weight-regularization. The computation in these approaches is challenging and so far have only been applied to small networks.

A more serious problem with weight regularization is its ineffectiveness in DNNs. For neural networks, the network outputs depend on the weights in a complex way and the exact value of the weights may not matter much. What ultimately matters is the network outputs (Benjamin et al., 2018), and it is better to directly regularize those.

**Deep Q-learning algorithms**   Deep Q-network (**DQN**) (Mnih et al., 2013) has been the first DQL algorithm to solve Atari games from images. To learn the $Q$-values, DQN minimizes the squared

Bellman error where the target $Q$-values are estimated by a target network with a periodically updated set of parameters. **Polyak DQN** is a DQL algorithm that also minimizes the squared Bellman error but uses a target network using a moving average of the parameters. **Double DQN** decouples the action selection from the action evaluation by using different network to do both and to provide a target during training and counter the maximum bias. It has been shown an effective way to stabilize DQN and prevent (soft-)divergence (Van Hasselt et al., 2016).

## 3 DEEP $Q$-LEARNING WITH FUNCTIONAL REGULARIZATION

In practice, deep $Q$-learning methods are known to be unstable, to not approximate the true value function, and to sometimes even (soft-)diverge (Van Hasselt, 2010; Van Hasselt et al., 2016; 2018). A multitude of solutions have been proposed to stabilize training, but in this work we will mainly focus on the stabilizing role played by target networks.

Target networks are parametrized by an additional set of lagging parameters. Intuitively, since the lagging parameters are used to estimate the target $Q$-value, they can be interpreted as a form of implicit *prior*, preventing the $Q$-value estimates from changing too quickly. In this section, we use this insight to introduce the functional regularized squared Bellman error, a regularized loss to stabilize training while using up-to-date target values. In this process, we help formalize the notion of target networks as forms of functional priors.

### 3.1 FUNCTIONALLY REGULARIZED SQUARED BELLMAN ERROR

Target networks have coupled roles: providing target $Q$-values used for learning, and stabilizing training. We can decouple these two roles by using up-to-date parameters to estimate the target $Q$-values, and using a functional prior to regularize the $Q$-values. Following Benjamin et al. (2018), we proposed using an $L_2$ functional regularizer (FR) which penalizes the divergence between the current estimate of Q-value function $Q_{\boldsymbol{\theta}}(\mathbf{s}, \mathbf{a})$ and a lagging version of it $Q_{\overline{\boldsymbol{\theta}}}(\mathbf{s}, \mathbf{a})$. This gives us the following loss function:

$$l^{\text{FR}}(\boldsymbol{\theta}) = \frac{1}{2}\bigg( R(\mathbf{s}_t, \mathbf{a}_t) + \texttt{stop\_grad}\left(\gamma\mathbb{E}\left[Q_{\boldsymbol{\theta}}(\mathbf{s}_{t+1}, \mathbf{a}_{t+1})\right]\right) - Q_{\boldsymbol{\theta}}(\mathbf{s}_t, \mathbf{a}_t)\bigg)^2 + \tag{5}$$

$$\frac{\kappa}{2}\bigg( Q_{\boldsymbol{\theta}}(\mathbf{s}_t, \mathbf{a}_t) - Q_{\overline{\boldsymbol{\theta}}}(\mathbf{s}_t, \mathbf{a}_t)\bigg)^2,$$

where the up-to-date parameter $\boldsymbol{\theta}$ is used in the Squared Bellman Error along with a FR loss to stabilize $\boldsymbol{\theta}$ by regularizing the current state-action pair only. The expectation above is taken over the transition and policy, and $\kappa > 0$ is the regularization parameter. By $\texttt{stop\_grad}$, we indicate a function that prevents gradients from propagating into the target $Q$-value estimate.

Critically, unlike Eq. (4), the target $Q$-value estimates are supplied by the up-to-date $Q$-network, with the functional prior now separately serving to stabilize the estimate. This decoupling allows us to quickly propagate newly encountered rewards and maintain stability. As with a target network, we can update the functional prior periodically to control the stability of the $Q$-value estimates. Overall, $l^{\text{FR}}$ is arguably similar in complexity to $l^{\text{target}}$, requiring only an additional function evaluation for $Q_{\boldsymbol{\theta}}(\mathbf{s}_{t+1}, \mathbf{a}_{t+1})$ and an additional hyper-parameter, $\kappa$, which was not difficult to tune in all of our experiments (see Section 4). As will be discussed in the subsequent Section 3.2, target networks implicitly make a fixed trade-off between speed and stability, whereas $\kappa$ in the FR squared Bellman error provides additional flexibility and can be task dependent.

The functional regularization between $Q_{\boldsymbol{\theta}}$ and $Q_{\bar{\boldsymbol{\theta}}}$ can be seen as an approximation to the Kullback-Leibler (KL) divergence between two Gaussian processes where we assume identity covariance for these two processes. Since the covariance matrices might not be the identity in practice, further improvement might be possible by estimating the covariances at the cost of additional computation, e.g. (Pan et al., 2020). This interpretation draws a link with other popular regularization methods in policy optimization inspired by trust-region algorithms (Schulman et al., 2015a; Abdolmaleki et al., 2018b;a) where the quantity of interest, the policy, is KL regularized towards its past values. In the same spirit, for value-based methods, the FR Squared Bellman Error regularizes the $Q$ function estimates towards their past values.

## 3.2 Interpreting Target Networks through Functional Regularization

Given the similar regularization role accomplished by target networks and functional priors, it is reasonable to assume that target networks can be interpreted through the lens of FR. To validate this assumption, we compared the gradients, $\nabla_{\boldsymbol{\theta}} l(\boldsymbol{\theta})$, arising from each case (Eq. (4) & Eq. (5)) which allowed us to see how these learning procedures differed. For a complete derivation, see Appendix A. To simplify notation, we define the $Q$-value estimate at time $t$ as $Q_{\boldsymbol{\theta}}^{(t)} \equiv Q_{\boldsymbol{\theta}}(\mathbf{s}_t, \mathbf{a}_t)$, the corresponding expected change in the $Q$-value estimate as $\Delta Q_{\boldsymbol{\theta}}^{(t)} \equiv \mathbb{E}_{P,\pi}[Q_{\boldsymbol{\theta}}^{(t+1)}|\mathbf{s}_t, \mathbf{a}_t] - Q_{\boldsymbol{\theta}}^{(t)}$, and the reward as $R^{(t)} \equiv R(\mathbf{s}_t, \mathbf{a}_t)$. Then, for a given state-action pair, we can compare the two gradients:

$$\nabla_{\boldsymbol{\theta}} l^{\text{FR}}(\boldsymbol{\theta}) = -\big(R^{(t)} + \gamma(Q_{\boldsymbol{\theta}}^{(t)} + \underline{\Delta Q_{\boldsymbol{\theta}}^{(t)}}) - Q_{\boldsymbol{\theta}}^{(t)}\big)\nabla_{\boldsymbol{\theta}} Q_{\boldsymbol{\theta}}^{(t)} + \underline{\kappa}\big(Q_{\bar{\boldsymbol{\theta}}}^{(t)} - Q_{\boldsymbol{\theta}}^{(t)}\big)\nabla_{\boldsymbol{\theta}} Q_{\boldsymbol{\theta}}^{(t)}. \quad (6)$$

Replacing $\underline{\Delta Q_{\boldsymbol{\theta}}^{(t)}}$ with $\Delta Q_{\bar{\boldsymbol{\theta}}}^{(t)}$ and $\underline{\kappa}$ with $\gamma$ we recover the gradient of $l^{\text{target}}(\boldsymbol{\theta})$:

$$\nabla_{\boldsymbol{\theta}} l^{\text{target}}(\boldsymbol{\theta}) = -\big(R^{(t)} + \gamma(Q_{\boldsymbol{\theta}}^{(t)} + \underline{\Delta Q_{\bar{\boldsymbol{\theta}}}^{(t)}}) - Q_{\boldsymbol{\theta}}^{(t)}\big)\nabla_{\boldsymbol{\theta}} Q_{\boldsymbol{\theta}}^{(t)} + \underline{\gamma}\big(Q_{\bar{\boldsymbol{\theta}}}^{(t)} - Q_{\boldsymbol{\theta}}^{(t)}\big)\nabla_{\boldsymbol{\theta}} Q_{\boldsymbol{\theta}}^{(t)}. \quad (7)$$

By comparing the gradients, we see that, despite the differences in their loss formulations, target networks and FR result in nearly identical gradients, with two key exceptions. The first distinction, underlined in orange, is the estimated change in the $Q$-value. With target networks, this term is supplied by the lagging network parameters, $\bar{\boldsymbol{\theta}}$, and is therefore not up-to-date, whereas with FR, it is supplied by the up-to-date parameters. Thus, illustrating why FR can propagate updated value estimates faster. The second distinction, underlined in blue, is the weighting on the difference between the lagging and up-to-date $Q$-value estimates. With target networks, this is the discount factor, $\gamma$, whereas with FR, this is a separate hyper-parameter. $\gamma$ is usually part of the problem definition and thus cannot be tuned for a specific task. Using the gradient in Eq. (7), and defining $\hat{Q}_{\boldsymbol{\theta},\bar{\boldsymbol{\theta}}}^{(t)} \equiv Q_{\boldsymbol{\theta}}^{(t)} + \Delta Q_{\bar{\boldsymbol{\theta}}}^{(t)}$, we can derive a loss function, $\widetilde{l}^{\text{target}}$, that yields equivalent gradients as $l^{\text{target}}$, but is written in the FR form:

$$\widetilde{l}^{\text{target}}(\boldsymbol{\theta}) = \frac{1}{2}\bigg(R(\mathbf{s}_t, \mathbf{a}_t) + \texttt{stop\_grad}\Big(\gamma\mathbb{E}\Big[\hat{Q}_{\boldsymbol{\theta},\bar{\boldsymbol{\theta}}}(\mathbf{s}_{t+1}, \mathbf{a}_{t+1})\Big]\Big) - Q_{\boldsymbol{\theta}}(\mathbf{s}_t, \mathbf{a}_t)\bigg)^2 +$$
$$\frac{\gamma}{2}\bigg(Q_{\boldsymbol{\theta}}(\mathbf{s}_t, \mathbf{a}_t) - Q_{\bar{\boldsymbol{\theta}}}(\mathbf{s}_t, \mathbf{a}_t)\bigg)^2.$$

From this perspective, target networks effectively perform a special, restricted form of Gaussian FR, with $p(Q_{\boldsymbol{\theta}}) = \mathcal{N}(Q_{\bar{\boldsymbol{\theta}}}, \gamma^{-1})$, i.e., a precision weight that is not under direct control. In contrast, $l^{\text{FR}}$ allows us to separately adjust this weight to trade-off between stability and learning speed. Overall, this derivation illustrates the similarities between the squared Bellman error with target networks and the FR squared Bellman error. It also highlights the advantages of using the FR squared Bellman error over square Bellman error combined with target networks: up-to-date target $Q$-value estimates and control over the regularization through $\kappa$.

## 3.3 Comparison with Polyak updating

Polyak's updating is a common technique used to trade-off between speed and stability for control problems. Interestingly, if we assume the target network is an estimate in weight space of the latest $Q$-network, we can show that Polyak's updates implicitly performs parameter regularization on the weight estimate. To better understand the implicit regularization, let us consider a situation in which after each gradient step (indexed by $i$), we identified the target network as the solution to the problem:

$$\bar{\boldsymbol{\theta}}_{i+1} = \min_{\bar{\boldsymbol{\theta}}} \frac{1}{2}\|\bar{\boldsymbol{\theta}} - \boldsymbol{\theta}_i\|^2 + \frac{1-\tau}{2\tau}\|\bar{\boldsymbol{\theta}} - \bar{\boldsymbol{\theta}}_i\|^2 \quad (8)$$

whereby $\boldsymbol{\theta}_i$ is the latest set of weights of the $Q$-network and $\bar{\boldsymbol{\theta}}_i$ is the previous instance of the target network, and $\tau$ is the exponential averaging step size. Minimizing $\bar{\boldsymbol{\theta}}$ is obtained by computing the gradient of the problem and equating it 0, leading to $\bar{\boldsymbol{\theta}}_{i+1} = (1-\tau)\boldsymbol{\theta}_i + \tau\bar{\boldsymbol{\theta}}$, which is exactly Polyak's updating. This derivation illustrates that Polyak's update is indeed a form of weight regularization

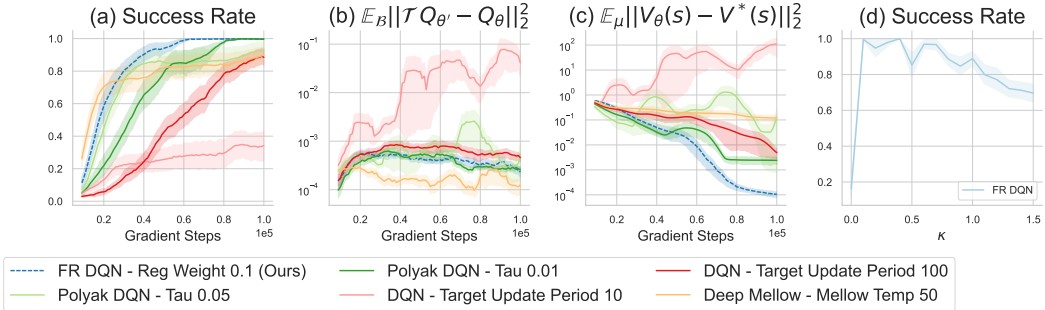

Figure 2: **Four Rooms Comparison**. We compare the performance of different Deep $Q$-learning algorithms. The reward position is kept fixed throughout training (upper-left room), but the agent's position is randomized at the beginning of every episode to bypass exploration difficulties. In Figure (a), we can see that FR DQN (blue dashed) quickly and stably completes the task. The other methods struggle to get a good combination of speed and stability, e.g., the two green methods are either fast or stable. All methods minimize the squared Bellman error equally well, as shown in Figure (b), however our method approximates the true value function an order of magnitude more accurately than the other methods as shown in Figure (c). Figure (d) shows the stability of FR DQN where the performance is not too sensitive to changes in the regularization parameter ($\kappa$).

with respect to the most recent target network weights. Weight regularization does not, however, guarantee that the output of the regularized network matches the previous target network. In fact, while this technique has found success in control problems (Lillicrap et al., 2015; Haarnoja et al., 2018a; Fujimoto et al., 2018), periodically updating the parameters is usually preferred for complex DNN architectures (Mnih et al., 2013; Hausknecht & Stone, 2015; Hessel et al., 2018; Kapturowski et al., 2018; Parisotto et al., 2020).

## 4 RESULTS

We compared the performance of our objective (FR-Squared Bellman Error) and other algorithms minimizing the squared Bellman error and using target networks in the Four Rooms environment and the subset of Atari games (Bellemare et al., 2013) used by Mnih et al. (2013). The experimental code is based on DQN Zoo (Quan & Ostrovski, 2020), implemented in Jax (Bradbury et al., 2018) and Haiku (Hennigan et al., 2020). Additional experimental details can be found in the Appendix.

### 4.1 FOUR ROOMS

#### 4.1.1 EXPERIMENTAL SET-UP

In order to better understand the performance of the FR squared Bellman error, we compared our objective to a suite of algorithms and hyper-parameters minimizing the squared Bellman error and using a target network in the Four Rooms environment (Sutton et al., 1999) with neural approximation. Four Rooms is a grid world environment comprised of four rooms divided by walls with a single gap in each wall allowing an agent to move between rooms. In this environment, the agent's must reach a goal state in the upper-left room near the right door, and the agent's initial position is drawn randomly from all other cells excluding the walls position. The agent's position is given through a one-hot encoding. The agent is capable of taking actions in four directions: up, down, left and right. See Fig. 12 in the Appendix for a graphical representation of the environment and the optimal discounted return.

The advantage of using this environment over more complex ones such as Atari games, is that the underlying transition matrix is available, and the true optimal value function can be computed using tabular methods (which will be referred to as $V^*$). We can then compare the agent's parametrized value estimate $V_{\theta}(\mathbf{s}) = \max_{\mathbf{a}} Q_{\theta}(\mathbf{s}, \mathbf{a})$ with the optimal one $V^*(\mathbf{s})$ to assess the accuracy of the agent's $Q$-values. The discount factor is 0.99, the reward is always 0 except at the goal state where every action leads to a reward of 1. Specifically all actions from the goal state receive a reward of

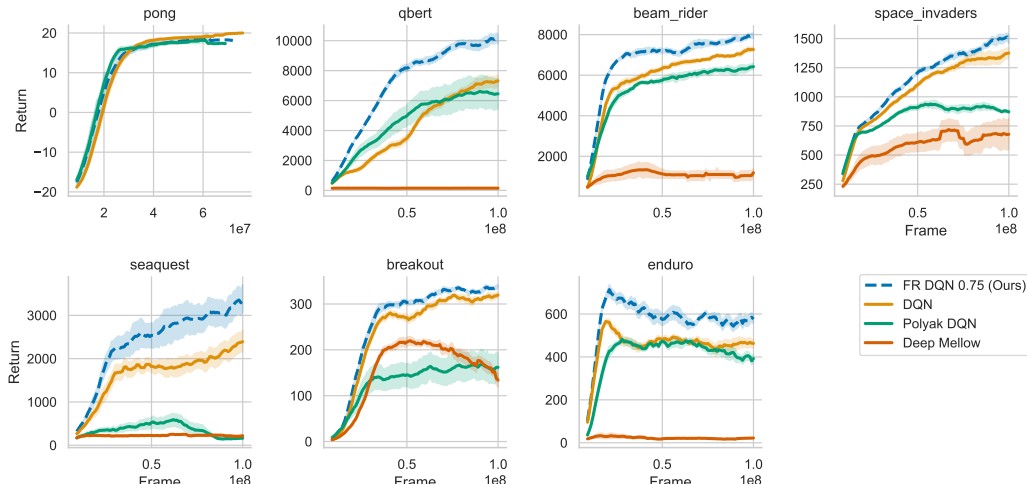

Figure 3: **DQN Atari Comparison**. Performance curves for the subset of Atari games from Mnih et al. (2013). The returns are averaged over 10 trials using the $\epsilon$-greedy policy with $\epsilon = 0.05$.

1. The episode terminates either once the agent reach the goal or after 500 steps. The environment is implemented using Easy MDP (Ahmed, 2020). The $Q$-value is parametrized by a Multi-Layers Perceptron (MLP) with 2 layers of 128 hidden units each. The shared hyper-parameters (learning rate and batch size) were tuned on 5 seeds for Deep $Q$-Network (DQN) (Mnih et al., 2013) to maximize success rate, and then use for all the other methods. The final performances are reported on 20 different seeds after 100k gradient steps (and as many transitions). All hyper parameters are available in Table 1.

### 4.1.2 RESULTS

We compared the performance of the FR square Bellman error combined with the DQN algorithm (FR DQN) to DQN, Polyak DQN, and Deep Mellow (Kim et al., 2019). As observed in Fig. 2 (a), the target network update period greatly affects performance of algorithms minimizing the squared Bellman error. On the one hand, frequently updating the target network leads to faster initial performance since the reward is propagated through the space faster. On the other hand, updating the network too frequently can lead to instability as observed in Fig. 2. For example, $\tau$ of 0.05 initially outperforms $\tau$ of 0.01, but becomes unstable, and never reaches optimal performance. The FR squared Bellman error combined with the DQN algorithm (FR DQN) bypasses this issue, by using the previous $Q$-values estimates exclusively for regularization which allows it to use the most up-to-date estimate for learning.

Additionally, $Q$-learning methods should converge to the optimal value function $V^*$, but in complex settings like Atari games, it is impossible to compute $V^*$ and monitor the distance to the optimum. In the Four Rooms setting, however, measuring the accuracy of their approximation is tractable. Interestingly, even if the Bellman error loss is roughly the same for FR DQN and Polyak DQN (Fig. 2 (b)), FR DQN approximates the true value function with much greater accuracy than Polyak DQN (Fig. 2 (c)). We also note that Deep Mellow does not recover the true value function. This failure is not surprising as Deep Mellow is biased by using the *mellowmax* operator instead of the *max*. This is of particular interest since the squared Bellman error is the metric that is usually monitored in practice and might not be an accurate measure of the accuracy of the $Q$-value.

### 4.2 ATARI

### 4.2.1 EXPERIMENTAL SET-UP

The Arcade Learning Environment (Bellemare et al., 2013) provides a set of benchmark Atari games for evaluating Reinforcement Learning algorithms. These games represent a good benchmark as

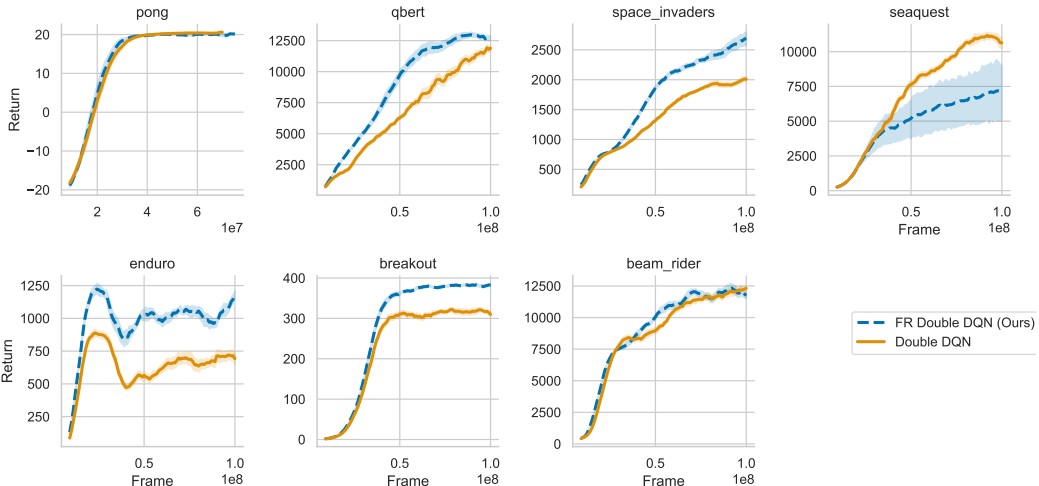

Figure 4: **Double DQN Atari Comparison**. Performance curves for the subset of Atari games from Mnih et al. (Mnih et al., 2013). The baseline (dotted blue line) and 4 different FR weights (solid lines) average returns over 10 trials using $\epsilon$-greedy with $\epsilon = 0.05$.

they possess rich visual observations and diverse reward functions. Complex DQL methods currently hold state-of-the-art results in this benchmark (Hessel et al., 2018). But in order to understand the effect of replacing the squared Bellman error with the FR squared Bellman error, we first compared our approach with the standard DQN (Mnih et al., 2013) and Polyak DQN. Then we showed that we can easily combine the FR squared Bellman error objective with orthogonal improvements such as Double DQN (Van Hasselt et al., 2016).

We stress that the algorithms based on the FR squared Bellman error, e.g., FR DQN and FR Double DQN, are simple modifications from the DQN Zoo library (Quan & Ostrovski, 2020). We tuned the hyper-parameter for each method (except DQN as they have already been tuned by the authors of the DQN Zoo package) by performing experiments on 2 seeds (only 1 set of global hyper-parameters used for every tasks). Once the best set of hyper-parameters for each method have been selected, we used 10 additional seeds (and exclude the 2 initial seeds) to compare the algorithms. All hyperparameters we used are displayed in Table 2 in the appendix. We used the same suite of representative environments from Mnih et al. (2013). Each seed took 48 hours to run on our resources for a total of 14400 hours of GPU usage.

### 4.2.2 RESULTS

**Comparison with DQN, Polyak DQN, and Deep Mellow** The comparison between FR DQN, DQN and Polyak DQN on Atari performance results are presented in Fig. 3. Across all environments, FR DQN matches or exceeds the performance of DQN and Polyak DQN in terms of sample efficiency and final performance at 100M steps. FR DQN's ability to use up-to-date target $Q$-values also plays a significant role in improving performance over Target Networks. Polyak DQN and Deep Mellow are quite unstable and do not constantly match the performance of DQN. We hypothesise that the temperature of Deep Mellow has to be finely tune to each games in order to outperform DQN as reported in Kim et al. (2019).

**Comparison with Double DQN** Since the introduction of Target Networks, there has been numerous techniques to stabilize the training and improve the performance of DQL methods (Hessel et al., 2018). While it is outside of the scope of this paper to combine FR DQN with each and every DQN improvements, we show that it is fairly easy to combine the FR squared Bellman error with double DQN (Van Hasselt et al., 2016). The results presented in Fig. 4 clearly demonstrates that the FR squared Bellman error can be combined with DDQN to match or achieves even higher performances on this subset of Atari games with the exception of Seaquest.

## 5    DISCUSSION

Multiple previous works have investigated how to improve value estimation through various constraints and regularization. Farahmand et al. (2009) is perhaps the closest work to our own, functionally regularizing the $L_2$ norm of the $Q$-value estimate, i.e., penalizing $\kappa||Q_{\boldsymbol{\theta}}||^2$. Farahmand et al. (2009)'s approach can be interpreted as using a fixed Gaussian prior, whereas we used a periodically updated set of parameters to provide a moving prior. However, penalising the magnitude of the $Q$-values would not prevent the algorithm to converge to $V^*$ if $\kappa$ does not tend towards 0. Shao et al. (2020) proposed adding an additional backward Squared Bellman Error loss to the next $Q$-value to stabilize training and remove the need for a target network. Other works have sought to regularize the $Q$-value estimator by constraining parameter updates, e.g., through regularization (Farebrother et al., 2018), conjugate gradient methods (Schulman et al., 2015b), pre-conditioning the gradient updates (Knight & Lerner, 2018; Achiam et al., 2019), or using Kalman filtering (Shashua & Mannor, 2019; 2020). As argued in Sections 2 and 3, functional regularization, as opposed to parameter regularization, is more appropriate in combination with DNNs, because parameter regularization does not necessarily imply that the $Q$-value estimates remain stable during training.

Other previous works have sought to address orthogonal issues with DQL algorithms, and, in principle, these could be combined with the FR squared Bellman error. For instance, Kim et al. (2019) removed the need for target network on Atari games by using the mellow max operator as an alternative to the max operator used in bootstrapping. This new operator is used with the squared Bellman error and could easily be used in combination with FR squared Bellman error. As mentionned above, Double Deep $Q$-learning (Van Hasselt, 2010; Van Hasselt et al., 2016) addresses positive bias in the value estimate (Thrun & Schwartz, 1993) by using two $Q$-networks, with each target network providing target values for the other. Fujimoto et al. (Fujimoto et al., 2018) extend and apply this technique to continuous control, instead using the minimum of the two target networks in the target value.

## 6    CONCLUSION

In this paper, we proposed the FR Squared Bellman Error as an alternative to the Squared Bellman Error. The Bellman error requires target networks to stabilize the training of DQL methods due to the non-stationary nature of the regression problem. Like the squared Bellman error combined with target networks, the FR squared Bellman error uses a lagging set of parameters to stabilize estimation. An important distinction however is that the lagging set is decoupled from target value estimation, instead taking the form of a functional prior. In this way, we can 1) use up-to-date target value estimates, thereby propagating reward information more quickly, and 2) separately control the degree of stabilization via the FR weight, $\kappa$. Indeed, by comparing the gradients resulting from the squared Bellman error with target networks to the FR squared Bellman error, we noted that target networks implicitly perform a form of FR, but with lagging target values and a non-independent weight ($\kappa = \gamma$). This helps illuminate why target networks stabilize training in practice, while also highlighting their drawbacks.

We also demonstrated the stability of FR DQN in the Four Rooms environment by showing that it can quickly recover the optimal value function, and show that we achieve higher performance than DQN and Polyak DQN. Then we showed that FR DQN consistently match or outperform DQN and Polyak DQN on a subset of the Atari suite. Finally we showed that FR DQN can be combined with DQN to further improve performance. Thus, we conclude that FR Bellman error provides a conceptually simple, easy to implement, and effective objective for improving DQL methods.

While the FR Bellman error requires an additional weighting parameter, $\kappa$, we noted that the performance does not change dramatically for small change in $\kappa$. We see $\kappa$ as providing needed flexibility in setting the degree of regularization, unlike Target Networks, which cannot separately control this aspect. In our experiments, we used a fixed $\kappa$ throughout training, however, future work could investigate methods for automatically adjusting $\kappa$ during training (Haarnoja et al., 2018b) or conditioning it on individual state-action pairs. Nevertheless, the FR Squared Bellman Error objective that we have presented here performs well across a wide range of environments, providing a drop-in replacement for the Squared Bellman Error.

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

## A APPENDIX

### A.1 $Q$-LEARNING WITH TARGET NETWORK

Letting $R^{(t)} \equiv R(\mathbf{s}_t, \mathbf{a}_t)$, $Q^{(t)} \equiv Q(\mathbf{s}_t, \mathbf{a}_t)$ and $\Delta Q_{\bar{\boldsymbol{\theta}}}^{(t)} \equiv \mathbb{E}[Q_{\bar{\boldsymbol{\theta}}}^{(t+1)} - Q_{\bar{\boldsymbol{\theta}}}^{(t)}|\mathbf{s}_t, \mathbf{a}_t] = \mathbb{E}[Q_{\bar{\boldsymbol{\theta}}}^{(t+1)}|\mathbf{s}_t, \mathbf{a}_t] - Q_{\bar{\boldsymbol{\theta}}}^{(t)}$

$$
\begin{aligned}
\nabla_{\boldsymbol{\theta}} l^{\text{target}}(\boldsymbol{\theta}) &= -\big(R^{(t)} + \gamma \mathbb{E}_{P,\pi}[Q_{\bar{\boldsymbol{\theta}}}^{(t+1)}] - Q_{\boldsymbol{\theta}}^{(t)}\big)\nabla_{\boldsymbol{\theta}} Q_{\boldsymbol{\theta}}^{(t)} \\
&= -\big(R^{(t)} + \gamma(Q_{\bar{\boldsymbol{\theta}}}^{(t)} + \Delta Q_{\bar{\boldsymbol{\theta}}}^{(t)}) - Q_{\boldsymbol{\theta}}^{(t)}\big)\nabla_{\boldsymbol{\theta}} Q_{\boldsymbol{\theta}}^{(t)} \\
&= -\big(R^{(t)} + \gamma(Q_{\bar{\boldsymbol{\theta}}}^{(t)} + \Delta Q_{\bar{\boldsymbol{\theta}}}^{(t)}) - (1 - \gamma + \gamma)Q_{\boldsymbol{\theta}}^{(t)}\big)\nabla_{\boldsymbol{\theta}} Q_{\boldsymbol{\theta}}^{(t)} \\
&= -\big(R^{(t)} + \gamma Q_{\bar{\boldsymbol{\theta}}}^{(t)} + \gamma\Delta Q_{\bar{\boldsymbol{\theta}}}^{(t)} - (1 - \gamma)Q_{\boldsymbol{\theta}}^{(t)} - \gamma Q_{\boldsymbol{\theta}}^{(t)}\big)\nabla_{\boldsymbol{\theta}} Q_{\boldsymbol{\theta}}^{(t)} \\
&= -\big(R^{(t)} + \gamma\Delta Q_{\bar{\boldsymbol{\theta}}}^{(t)} - (1 - \gamma)Q_{\boldsymbol{\theta}}^{(t)}\big)\nabla_{\boldsymbol{\theta}} Q_{\boldsymbol{\theta}}^{(t)} + \gamma\big(Q_{\bar{\boldsymbol{\theta}}}^{(t)} - Q_{\boldsymbol{\theta}}^{t}\big)\nabla_{\boldsymbol{\theta}} Q_{\boldsymbol{\theta}}^{(t)} \\
&= -\big(R^{(t)} + \gamma\Delta Q_{\bar{\boldsymbol{\theta}}}^{(t)} - Q_{\boldsymbol{\theta}}^{(t)} + \gamma Q_{\boldsymbol{\theta}}^{(t)}\big)\nabla_{\boldsymbol{\theta}} Q_{\boldsymbol{\theta}}^{(t)} + \gamma\big(Q_{\bar{\boldsymbol{\theta}}}^{(t)} - Q_{\boldsymbol{\theta}}^{(t)}\big)\nabla_{\boldsymbol{\theta}} Q_{\boldsymbol{\theta}}^{(t)} \\
&= -\big(R^{(t)} + \gamma(Q_{\boldsymbol{\theta}}^{(t)} + \underline{\Delta Q_{\bar{\boldsymbol{\theta}}}^{(t)}}) - Q_{\boldsymbol{\theta}}^{(t)}\big)\nabla_{\boldsymbol{\theta}} Q_{\boldsymbol{\theta}}^{(t)} + \underline{\gamma}\big(Q_{\bar{\boldsymbol{\theta}}}^{(t)} - Q_{\boldsymbol{\theta}}^{(t)}\big)\nabla_{\boldsymbol{\theta}} Q_{\boldsymbol{\theta}}^{(t)}
\end{aligned}
$$

### A.2 $Q$-LEARNING WITH FUNCTIONAL REGULARIZATION

$$
\begin{aligned}
\nabla_{\boldsymbol{\theta}} l^{\text{FR}}(\boldsymbol{\theta}) &= -\big(R^{(t)} + \gamma \mathbb{E}_{P,\pi}[Q_{\boldsymbol{\theta}}^{(t+1)}] - Q_{\boldsymbol{\theta}}^{(t)}\big)\nabla_{\boldsymbol{\theta}} Q_{\boldsymbol{\theta}}^{(t)} + \kappa\big(Q_{\bar{\boldsymbol{\theta}}}^{t} - Q_{\boldsymbol{\theta}}^{(t)}\big)\nabla Q_{\boldsymbol{\theta}}^{(t)} \\
&= -\big(R^{(t)} + \gamma(Q_{\boldsymbol{\theta}}^{(t)} + \underline{\Delta Q_{\boldsymbol{\theta}}^{(t)}}) - Q_{\boldsymbol{\theta}}^{(t)}\big)\nabla_{\boldsymbol{\theta}} Q_{\boldsymbol{\theta}}^{(t)} + \underline{\kappa}\big(Q_{\bar{\boldsymbol{\theta}}}^{(t)} - Q_{\boldsymbol{\theta}}^{t}\big)\nabla Q_{\boldsymbol{\theta}}^{(t)}
\end{aligned}
$$

### A.3 CONVERGENCE OF FUNCTIONALLY REGULARIZED VALUE ITERATION

Let us consider the value iteration algorithm for the state-action value function in the tabular setting.

$$Q_{t+1} \leftarrow \mathcal{T}^{\pi} Q_t$$

where $\mathcal{T}^{\pi}(Q) = R + \gamma P^{\pi} Q$ It is known that this algorithm converges linearly as this update is a contraction. We show in this section that value iteration with functional regularization enjoys similar properties.

**Theorem 1** (Convergence of functionally regularized value iteration). *Value iteration with learning rate $\eta$ and with functional regularization $\ell(Q, \bar{Q}) = \frac{\kappa}{2}\|Q - \bar{Q}\|^2$ where $\bar{Q}$ is updated every $T$ steps to the current $Q$ value converges linearly for $\kappa \geq 0$, $\eta \leq \frac{1}{1+\gamma+\kappa}$ and $T > \frac{\log(1+2\frac{\kappa}{1-\gamma})}{\log\frac{1}{\rho}}$.*

*Proof.* For this proof, we need to show that $Q$ will converge to $Q^\pi$. For this we split the problem in two 1) on one hand we show that $Q$ converges to $Q_\kappa^\pi(\bar{Q})$, the best FR-regularized $Q$ value and 2) that $Q_\kappa^\pi(\bar{Q})$ converges to $Q^\pi$ depending on $\bar{Q}$.

**First step $Q$ and $Q_\kappa^\pi(\bar{Q})$:** Our loss function is

$$\tfrac{1}{2}\|\mathcal{T}^\pi(Q) - Q\|^2 + \tfrac{\kappa}{2}\|Q - \bar{Q}\|^2$$

where $\bar{Q}$ is a prior state-action value function that will be update every $T$ steps.

The update for FR-regularized loss would be

$$Q_{t+1} \leftarrow Q_t + \eta\big(\mathcal{T}^\pi Q_t - Q_t - \kappa(Q_t - \bar{Q})\big)$$

Therefore, equating $Q_t = Q_{t+1}$, the fixed point is

$$Q_\kappa^\pi(\bar{Q}) = ((1+\kappa)I - \gamma P^\pi)^{-1}(R + \kappa\bar{Q})$$

Let us look at the Euclidean distance

$$
\begin{aligned}
\|Q_{t+1} - Q_\kappa^\pi(\bar{Q})\| &= \|\big(I - \eta(\gamma P^\pi - (1+\kappa)I)\big)Q_t - Q_\kappa^\pi(\bar{Q})\| \\
&\leq \rho\|Q_t - Q_\kappa^\pi(\bar{Q})\|
\end{aligned}
$$

This is guaranteed to be a contraction for the spectral radius $\rho = \max_{\|x\|=1} x^\top\big(I - \eta((1+\kappa)I - \gamma P^\pi)\big)x < 1$. As the spectral radius of $P^\pi$ is 1 (Perron-Frobenius theorem), we can deduce that $\eta < \frac{2}{1+\gamma+\kappa}$ is sufficient to get the spectral radius $I - \eta((1+\kappa)I - \gamma P^\pi)$ bounded by 1.

Thus we have

$$\|Q_{T+t} - Q_\kappa^\pi(Q_t)\| \leq \rho^T\|Q_t - Q_\kappa^\pi(Q_t)\|$$

**Second step $Q_\kappa^\pi(\bar{Q})$ and $Q^\pi$:** First let us show that $Q_\kappa^\pi(\bar{Q}) = Q^\pi + (I + \kappa - \gamma P^\pi)^{-1}\kappa(\bar{Q} - Q^\pi)$

We have $Q_\kappa^\pi(\bar{Q}) = ((1+\kappa)I - \gamma P^\pi)^{-1}(R + \kappa\bar{Q})$ and $Q^\pi = (I - \gamma P^\pi)^{-1}R$. Using $A^{-1} - B^{-1} = A^{-1}(B - A)B^{-1}$, we get

$$
\begin{aligned}
Q_\kappa^\pi(\bar{Q}) - Q^\pi &= ((1+\kappa)I - \gamma P^\pi)^{-1}(-\kappa)(I - \gamma P^\pi)^{-1}R + ((1+\kappa)I - \gamma P^\pi)^{-1}\kappa\bar{Q} \\
&= (I + \kappa - \gamma P^\pi)^{-1}\kappa(\bar{Q} - Q^\pi)
\end{aligned}
$$

**Putting it all together:** Let us define the FR Bellman operator $Q \mapsto \mathcal{T}_{FR}^T[Q]$ which consists into updating the prior in FR to the current $Q$-value then doing $T$ steps of FR update on $Q$. We have $\mathcal{T}_{FR}^T[Q^\pi] = Q^\pi$ as $Q^\pi = R + \gamma P^\pi Q^\pi$.

$$
\begin{aligned}
\|\mathcal{T}_{FR}^T[Q_t] - Q^\pi\| &\leq \|Q_{t+T} - Q_\kappa^\pi(Q_t)\| + \|Q_\kappa^\pi(Q_t) - Q^\pi\| \\
&\leq \rho^T\|Q_t - Q_\kappa^\pi(Q_t)\| + \|(I + \kappa - \gamma P^\pi)^{-1}\kappa\|_2\|Q_t - Q^\pi\| \\
&\leq \rho^T\|\big(I - (I + \kappa - \gamma P^\pi)^{-1}\kappa\big)\big(Q^\pi - Q_t\big)\| + \mu\|Q_t - Q^\pi\| \\
&\leq \big(\rho^T(1+\mu) + \mu\big)\|Q_t - Q^\pi\|
\end{aligned}
$$

With $\mu = \text{Spectral radius}\big((I + \kappa - \gamma P^\pi)^{-1}\kappa\big) \leq \frac{\kappa}{1-\gamma+\kappa} < 1$ as $1 - \gamma > 0$. Thus, for

$$T > \frac{\log(1 + 2\frac{\kappa}{1-\gamma})}{\log\frac{1}{\rho}} \geq \frac{\log\frac{1+\mu}{1-\mu}}{\log\frac{1}{\rho}}$$

we have that $\big(\rho^T(1+\mu) + \mu\big) < 1$ therefore our operator is a contraction, thus the methods converges linearly.

$\square$

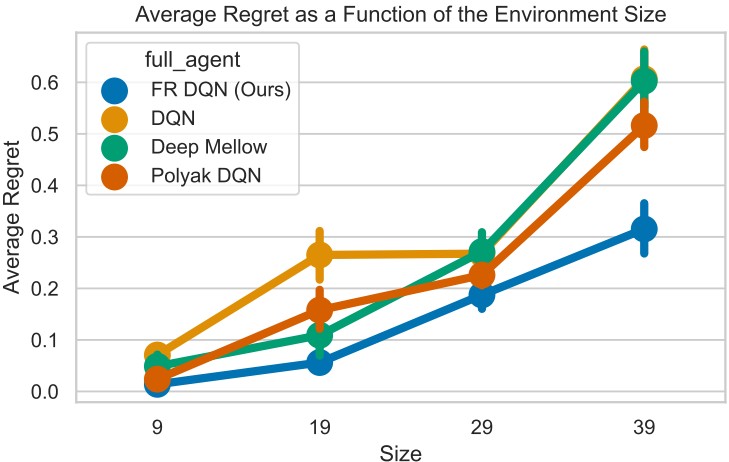

Figure 5: We benchmark the different algorithms on different Four Room environment sizes. As we increase the size of the Four Room environment, the reward becomes sparser and the task more difficult to complete. We observe that FR DQN scale more gracefully to larger environment sizes. We report the average regret over 500k iterations. See Table 1 for the hyper-parameter used for each algorithm.

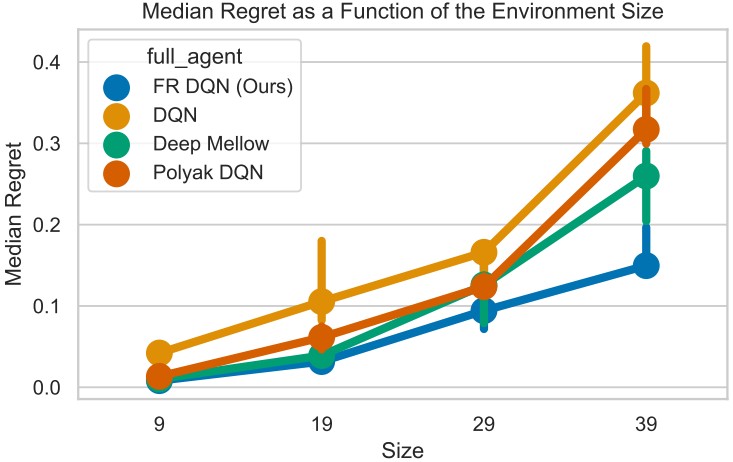

Figure 6: The average regret can be noisy since certain runs might not complete the task, e.g. 2 seeds do not complete the task for DQN on the size 19. Thus, we also report the median regret over 500k iterations. See Table 1 for the hyper-parameter used for each algorithm.

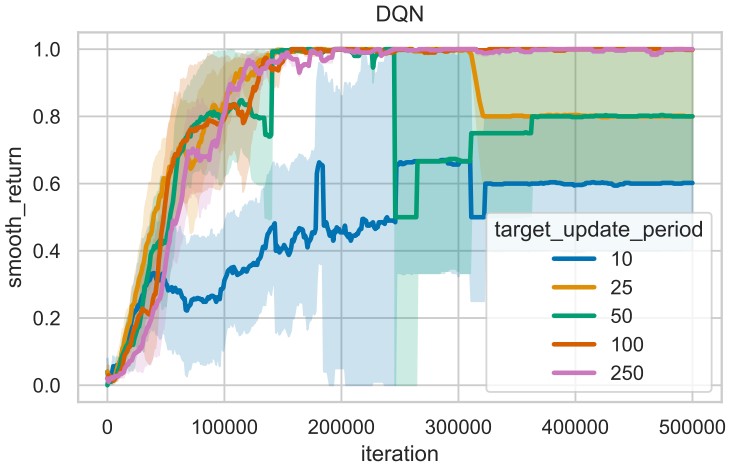

Figure 7: DQN with different target update period.

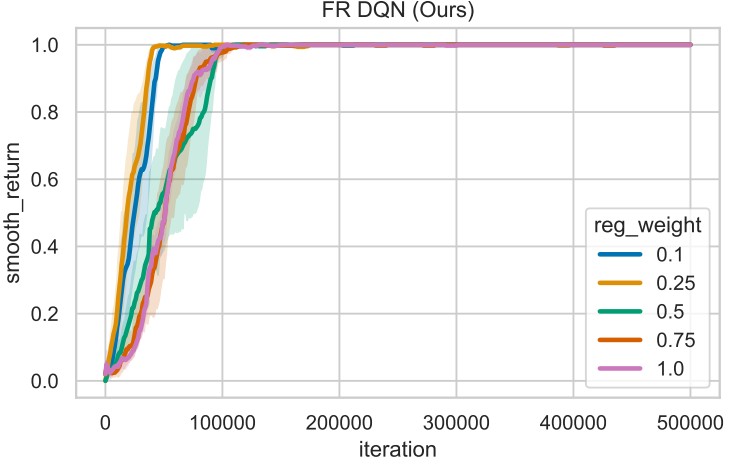

Figure 8: FR DQN with different regularization weight.

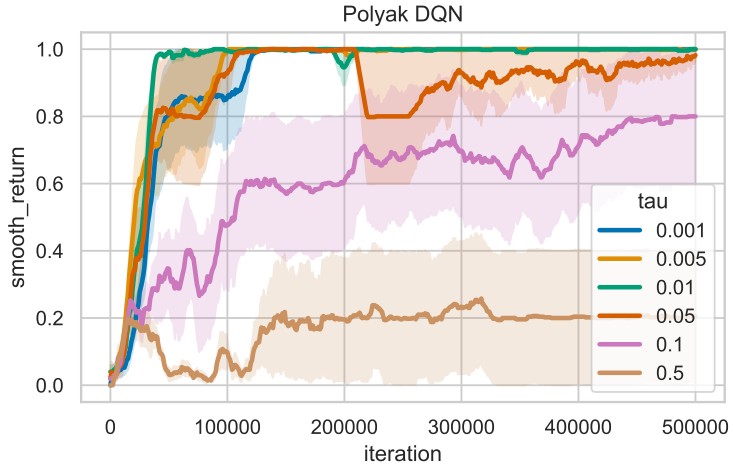

Figure 9: Polyak DQN with different parmaeter $\tau$.

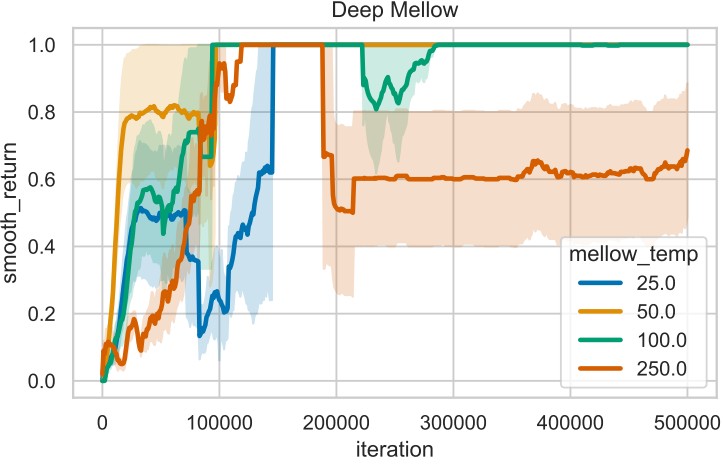

Figure 10: Deep Mellow with different temperature parameter.

Table 1: Four Rooms Hyperparameters

| Hyperparameter | DQN | Polyak DQN | FR DQN | Deep Mellow |
|---|---|---|---|---|
| Learning rate | 1e-3, **1e-4** | 1e-4 | 1e-4 | 1e-4 |
| Batch size | 32, **512** | 512 | 512 | 512 |
| Target Update Period | 10, 25, 50, **100**, 250 | 1 | - | - |
| Prior Update Period | - | - | 50, 500, **5000** | - |
| $\tau$ | - | 5e-1, 1e-1, 5e-2, **1e-2** | - | - |
| $\kappa$ | - | - | [**0.1**, ..., 0.9] | - |
| Mellow Temperature | - | - | - | 25, **50**, 100, 250 |
| Deep Neural Network | [128, 128] | [128, 128] | [128, 128] | [128, 128] |
| $\epsilon$ | 0.05 | 0.05 | 0.05 | 0.05 |

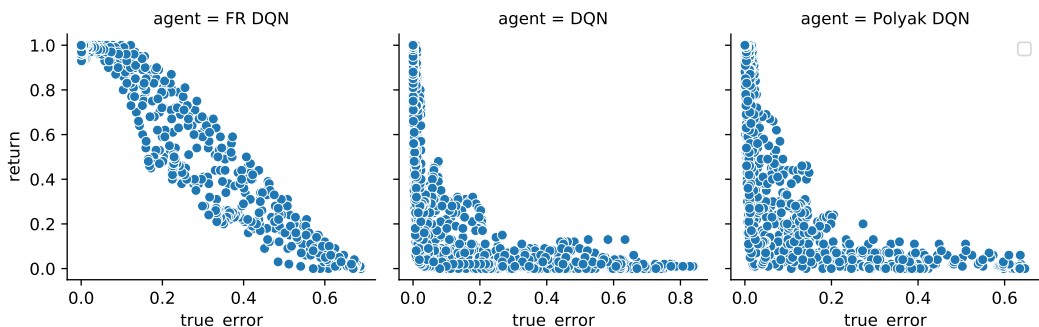

Figure 11: Return.

## B    APPENDIX

### B.1    FOUR ROOMS

### B.2    ATARI

Most of the hyper-parameters have been kept from the defaut DQN and Double DQN respectively. Here is a list of the tuning that has been performed and the resulting best hyper-parameters.

Table 2: Atari Hyperparameters

| Hyperparameter | Polyak DQN | FR DQN | FR Double DQN | Deep Mellow |
|---|---|---|---|---|
| $\tau$ | 5e-5, 2.5e-5, **1e-5**, 5e-6 | - | - | - |
| Target Update Period | 100 | - | - | - |
| Prior Update Period | - | 4e4 | 4e4 | - |
| $\kappa$ | - | 0.25, 0.5, **0.75** | **0.75** | - |
| Mellow Temperature | - | - | - | **25**, 50, 75, 100 |

## C    ADDITIONAL EXPERIMENTAL DETAILS

## D    RETURN AND OPTIMAL VALUE FUNCTION ERROR

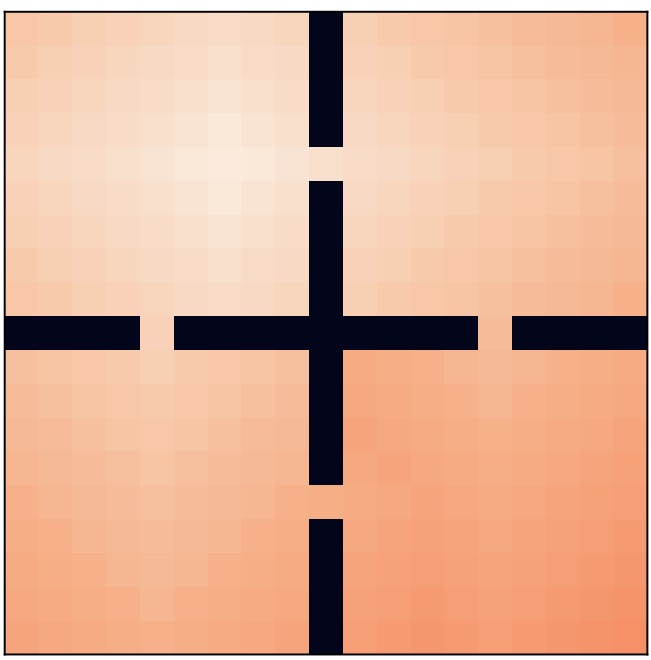

Figure 12: Value function of the 4 Rooms environments estimated via tabular $Q$-learning.

