# OpenReview forum: "Beyond Target Networks: Improving Deep $Q$-learning with Functional Regularization"
_ICLR.cc/2022/Conference — ICLR 2022 Submitted_

### Official Review · Reviewer_Gxgv · 2021-10-18

**Correctness:** 3
**Technical Novelty And Significance:** 2
**Empirical Novelty And Significance:** 2
**Recommendation:** 5
**Confidence:** 4

**Main Review:**

Overall the paper is well written and easy to follow. The idea of using the target network for “functional regularization” is simple (in a good sense) and well-motivated. The novelty is somewhat limited, but for empirical papers I think performance is more important than novelty. The authors include code that strengthens their arguments. Given that the paper presents no theorems, I think the sections which mathematically compare the proposed methods to previous ones could be made a little smaller. At any rate, my main concern is the empirical evaluation.

## Four rooms

The four rooms environment is a simple toy environment. It is reasonable to use such an environment as a sanity check to e.g. show that some theoretical property holds. However, the proposed method lacks theoretical grounding in the form of theorems or guarantees and thus relies on empirical testing to demonstrate any virtues. I think that the four rooms environment is too simple for any empirical arguments, and that performance on it shows very little for an empirical method.


## Atari

The first issue with the atari environment is the number of environments. The authors use 7 tasks considered by previous researchers and use 10 seeds for each of these. I think it would be better to use two seeds in each Atari environment (I think there are ~55).  This would not lead to statistically significant results on individual tasks, but I think it would lead to a better picture of the agents' performance once the authors average human-normalized scores across environments. Just using a few tasks raises the risk that the group of 7 tasks is cherry-picked (in the sense that there are many such groups to choose from) which undermines the results.

The second issue with the atari environment is the baselines. The DQN algorithm is over 5 years old at this time and has seen many improvements, so improving upon it does not advance the state-of-the-art. The authors state that “it is outside of the scope of this paper to combine FR DQN with each and every DQN improvement”. This is understandable, but what can be even more impactful is improving the performance of competitive methods. I would suggest that the authors consider ONE state-of-the-art baseline that is more competitive and show that their methods improve it. Suitable baselines might be spr (https://github.com/mila-iqia/spr) or rainbow (https://github.com/google/dopamine/tree/master/dopamine/agents/rainbow), both of which have excellent open-sourced code-bases. For such methods, the authors would also not need any parameter tuning.


**Summary Of The Paper:**

In Q-learning, a target network is often used to define the Bellman loss. The authors propose to estimate the Bellman loss with the “online” network and use the target network for functional regularization. In practice, functional regularization means minimizing the square difference between the target and the online network. Empirically, the authors show that this can improve performance over 7 Atari tasks and the four rooms environment.


**Summary Of The Review:**

The paper presents a straightforward and well-motivated idea for using online q-values for calculating the bellman loss. The paper is well written and easy to follow. Since there are essentially no theoretical contributions, the empirical evaluation is important but unfortunately has several problems. The authors consider the toy environment of four rooms, which I believe is too simple. The authors also consider 7 Atari environments, I think it would be better to consider all environments from the Atari suite. Most importantly, for the Atari environments, the baselines considered are very simple, and improving upon them does not show that the state-of-the-art is being advanced.

---

> ### Author Response · Authors · 2021-11-19
> **Response to Reviewer Gxgv**
>
> Thank you for your comments and suggestions.
>
> **The novelty is somewhat limited, but for empirical papers I think performance is more important than novelty. The authors include code that strengthens their arguments.**
>
> To our knowledge, it is the first time that FR has been used to regularize value estimation in RL. This perspective allows us to have a novel and coherent view of target networks. Thus, even if the FR techniques have been used elsewhere, we believe that the perspective of framing target networks as functional regularization is novel and could lead to a myriad of new works in this direction.
>
> **Given that the paper presents no theorems, I think the sections which mathematically compare the proposed methods to previous ones could be made a little smaller.**
>
> We added a theorem for functionally regularized value iteration and a proof of convergence in the appendix. See the general response.
>
> **I think that the four rooms environment is too simple for any empirical arguments, and that performance on it shows very little for an empirical method.**
>
> The purpose of the Four Room environment is not to showcase state-of-the-art performance, but to help the reader and investigate phenomena that cannot be investigated on Atari, e.g. convergence to Q*. We found these didactic examples to be very common and helpful in the field. To further improve the didactic nature of the Four Room environment, we have added comparisons for different sizes. Increasing the size of the environment makes the reward sparser. The results are available in Figure 5 in the appendix and show that FR DQN scale gracefully as we increase the size of the environment.
>
> **The first issue with the atari environment is the number of environments. The authors use 7 tasks considered by previous researchers and use 10 seeds for each of these. I think it would be better to use two seeds in each Atari environment (I think there are ~55). This would not lead to statistically significant results on individual tasks, but I think it would lead to a better picture of the agents' performance once the authors average human-normalized scores across environments. Just using a few tasks raises the risk that the group of 7 tasks is cherry-picked (in the sense that there are many such groups to choose from) which undermines the results.**
>
> We want to stress that we used these 7 games because they were the same used by the initial DQN paper and were carefully chosen to benchmark their methods. We did not try our algorithm on any other Atari games.
> Moreover, the variance on some of these tasks is enormous, see https://raw.githubusercontent.com/deepmind/dqn_zoo/master/plot_atari_individual.svg (see freeway, chopper command, bank heist, seaquest, etc.), which would make the comparison on these games meaningless for only 2 seeds. Thus, we believe that FR combined with 2 algorithms evaluated with 10 seeds on 7 tasks chosen by a third party (Mnih 2013) is plenty of evidence to judge our method.
>
> **The second issue with the atari environment is the baselines. The DQN algorithm is over 5 years old at this time and has seen many improvements, so improving upon it does not advance the state-of-the-art. The authors state that “it is outside of the scope of this paper to combine FR DQN with each and every DQN improvement”. This is understandable, but what is most relevant for a conference like ICLR is improving the performance of competitive methods. I would suggest that the authors consider ONE state-of-the-art baseline that is more competitive and show that their methods improve it. Suitable baselines might be spr (https://github.com/mila-iqia/spr) or rainbow (https://github.com/google/dopamine/tree/master/dopamine/agents/rainbow), both of which have excellent open-sourced code-bases. For such methods, the authors would also not need any parameter tuning.**
>
> The goal of the paper is to explore how target networks relate to regularization methods. We then show that explicitly regularizing the Q value estimate improves several baselines. We think it is better that the method works on several reasonable baselines (DQN and double DQN) rather than just one sota method. We also disagree with the reviewer that the goal of the conference is to only improve competitive methods. The goal is to improve our scientific knowledge and understanding of current methods, which our paper does. Rainbow is a combination of many techniques which adds a lot of complexity, interactions, and confounding variables.

---

> > ### Comment · Reviewer_Gxgv · 2021-11-19
> > **comment**
> >
> > I thank the authors for their response.
> >
> > **We think it is better that the method works on several reasonable baselines (DQN and double DQN) rather than just one sota method.**
> >
> > A potential problem here is that you are just reinventing the wheel. Perhaps the issue you "solve" is already solved by e.g. distributional RL, or by dueling q-learning, etc. If everyone keeps on improving the same baseline without comparing to improvements beyond the baseline, the RL field might not see any "real" progress as measured by the score of the best-performing agent.
> >
> > **We also disagree with the reviewer that the goal of the conference is to only improve competitive methods.**
> >
> > This is not a belief I have. For a paper such as the original submission, which mainly proposes a new algorithm that improves empirical performance I think advancing the SOTA is a very important goal. The original submission had few theoretical contributions, and I don't think the proposed idea is very novel or that it provides much new and deep understanding. For this reason, empirical evaluation becomes very important, and in my opinion, beating the SOTA is the strongest type of empirical result.

---

> > > ### Author Response · Authors · 2021-11-19
> > > **response**
> > >
> > > We would like to reiterate that we are not adding a new trick on top of DQN, we are investigating the role of target networks and replacing them with a more flexible and theoretically grounded alternative for which we have a tabular proof of convergence. We show that our more theoretically grounded regularization fulfills the role of the target networks (which Distributional RL, Duelling networks do not) while being more flexible and stable. The fact that distributional RL and dueling networks still use target networks makes it very clear that none of them tackle the issue we are addressing.

---

### Official Review · Reviewer_PwAu · 2021-10-25

**Correctness:** 2
**Technical Novelty And Significance:** 3
**Empirical Novelty And Significance:** 2
**Recommendation:** 5
**Confidence:** 4

**Main Review:**

Strength:
1. The idea of using up-to-date parameters in the target network looks interesting. It was believed that the target network needs to be periodically updated for stability, so this paper brings a new perspective on the target network in DQNs.

2. The authors showed empirically that the proposed method can outperform DQN baselines across a few different domains. The performance gap indeed shows the promise of the proposed method.

Weakness:
1. It's not convincing to me that the regularization term in Equation (5) can be interpreted as a KL divergence between two GPs. As the authors stated, "mean functions are the Q-values at different state-action pairs and whose covariance function is the identity". Given that Q_{\bar{\theta}} corresponds to a lagged version of Q_{\theta_t}, the assumption may not hold, since covariance functions are not necessarily the same and equal to the identity matrix.

2. As a part of the motivation, the authors mentioned a few times about the lack of propagating newly encountered rewards, due to the use of target networks. This statement is a bit problematic: Since rewards are randomly sampled from a replay buffer in DQNs, it's difficult to guarantee whether the used rewards can be newly encountered or not.

3. The experiments did show the advantages over DQN and DDQN baselines. However, it lacks the comparison against a very related work as follows, which also removes the need for target networks in DQNs. A direct comparison could strength this work empirically.

     Seungchan Kim, Kavosh Asadi, Michael Littman, and George Konidaris. Deepmellow: removing the need for a target network in deep q-learning. In Proceedings of the Twenty Eighth International Joint Conference on Artificial Intelligence, 2019.

**Summary Of The Paper:**

This paper proposed a new version of deep Q-learning, by avoiding the use of target networks in Bellman errors. Instead, the authors added a regularization term to enforce the current Q-value estimates close to its lagged version. Some analysis was provided to draw the connection with previous work on functional regularization and Polyak updating. Finally, the authors showed in experiments that the proposed method achieved better or similar performance, when compared with DQN and DDQN baselines.

**Summary Of The Review:**

This paper brings a new perspective for the use of target networks in DQNs, and the experiments show some promise. However, the interpretation on why it works has some flaws, and a more rigorous analysis could make it more convincing. Furthermore, the authors need to compare with the related work above.

---

> ### Author Response · Authors · 2021-11-19
> **Response to Reviewer PwAu**
>
> Thank you for your comments and suggestions.
>
> **Weakness 1. It's not convincing to me that the regularization term in Equation (5) can be interpreted as a KL divergence between two GPs. As the authors stated, "mean functions are the Q-values at different state-action pairs and whose covariance function is the identity". Given that Q_{\bar{\theta}} corresponds to a lagged version of Q_{\theta_t}, the assumption may not hold, since covariance functions are not necessarily the same and equal to the identity matrix.**
>
> An example of a KL between 2 GPs approximate with 2 DNNs can be seen in Equation 7 of https://arxiv.org/abs/2004.14070 . If you assume that K is the identity matrix you recover the KL minus a constant entropy term (as K is the identity) and end up with the square difference between the mean predictions of 2 DNNs. In our paper, we assume that K is the identity and we approximate the KL with the batch samples.
>
>  You are right, the assumption of the covariance matrices being equal to the identity might not hold in practice, but estimating them is expensive. Thus we perform an approximation by assuming that the covariances are the identity that could potentially be improved by modeling the covariances. This is an interesting direction for future work. Nonetheless, the text now reads:
> "The functional regularization between $Q_\vtheta$ and $Q_{\bar{\vtheta}}$ can be seen as an approximation to the Kullback-Leibler (KL) divergence between two Gaussian processes where we assume identity covariance for these two processes. Since the covariance matrices might not be the identity in practice, further improvement might be possible by estimating the covariances at the cost of additional computation, e.g. Pan el al. 2020."
>
>
> **Weakness 2. As a part of the motivation, the authors mentioned a few times about the lack of propagating newly encountered rewards, due to the use of target networks. This statement is a bit problematic: Since rewards are randomly sampled from a replay buffer in DQNs, it's difficult to guarantee whether the used rewards can be newly encountered or not.**
>
> We agree that the previous statement was a bit confusing. The point is that newly encountered rewards will still be used to evaluate the TD-loss. However, they will only be used to update the value at the current state, as the target network's output remains fixed within the update period. By removing the target network, newly encountered rewards can propagate to all previously encountered states (via target values from the current value network). The text now reads:  “Newly encountered reward used to evaluate the loss function won’t be propagated as long as the target network is not updated.”
>
> **Weakness 3. The experiments did show the advantages over DQN and DDQN baselines. However, it lacks the comparison against a very related work as follows, which also removes the need for target networks in DQNs. A direct comparison could strength this work empirically.
> Seungchan Kim, Kavosh Asadi, Michael Littman, and George Konidaris. Deepmellow: removing the need for a target network in deep q-learning. In Proceedings of the Twenty Eighth International Joint Conference on Artificial Intelligence, 2019.**
>
> We have included the requested baseline. See the general response.

---

### Official Review · Reviewer_yew3 · 2021-10-29

**Correctness:** 4
**Technical Novelty And Significance:** 3
**Empirical Novelty And Significance:** 3
**Recommendation:** 8
**Confidence:** 4

**Main Review:**

The paper is well-written and easy to follow along. I liked reading the paper as its motivation is clearly lined out from the beginning and the claims are also supported by a theoretical justification. Target networks often provide an easy solution to get an RL agent running but properly tuning their update frequency remains difficult in practice. This is also due to the fact that all their implications are largely still unknown. Functional regularization of the prediction of the network presents an easy solution, whose implication becomes clear through the provided theory. The proposed solution should also be (despite not being mentioned by the authors) much more computationally efficient than both approaches that use target networks and related approaches that try to remove the need for target networks. That being said, the authors could also highlight this a bit more.

From an experimental point of view, the used environments are sufficient. Atari serves well as a benchmarking environment and comparing the approach to DQN and DDQN also seems sufficient (first I wondered why the other improvements are not being used but those really should not affect each other, so it is better to keep it as simple as possible, DDQN makes sense; also, I wondered how results in DDPG should look like – however, the results should also hold here). Question: did you also use tuned DDQN hyperparameters from DQN Zoo for the experiments? This is not mentioned anywhere.

However, what I am missing a bit is an experimental benchmark against related approaches that have been proposed (e.g., Shao et al. (2020) and Kim et al. (2019) which have both been discussed as similar approaches in the paper). I don’t expect them to be much worse or much better either. The proposed functional regularization remains much more intuitive and tractable and also more computationally efficient (which at least for me would the killer-argument to used FR over other that trade performance in sample efficiency over ease of implementation).

Some minor comments:
-	p4: In practice, deep Q-learning methods are known to be unstable, [to] not approximate the true value function, and [to] sometimes even (soft-)diverge (Van Hasselt, 2010; Van Hasselt et al., 2016; 2018).
-	Sec. 4.1.2: for the experiments in Fig. 2(a): do other DQN-variants that use other target network update periods lie in between those two graphs (10 and 100)?
-	Sec. 4.1.2: \tau=0.05 initially outperforms \tau=0.01 […]. Does it reach the plateau in the end (looks like it will but not sure)
-	Sec. 4.2.1: “Complex DQL methods currently hold state-of-the-art results in this benchmark (Hessel et al., 2018). As previously mentioned, complex DQL methods show state-of-the-art results in this benchmark task.” – this seems redundant.
-	Can you explain why in Sequest everything is different?
-	Fig 2, d) Success rate is quite high for different kappa (in [0.7, 1.0]), but still somewhat “sensible”. In the last part of Sec. 6 Discussion they claim “[…] the performance does not dramatically change for small changes in kappa […].”; Also: It would have been nice to see convergence speed for different values of kappa (as opposed to only success rate)
-	p4: Typo “sectionSection 3.2”


**Summary Of The Paper:**

The paper proposes a functional regularization scheme to constrain updates to the Q-value estimates to get target networks removed. The paper is theoretically justified and puts the scheme into the context of delayed target networks and polyak-averaging (that both instead introduce a lag to the Q-values estimated by the neural network). The paper provides experiments on some Atari benchmarks against a DQN and DDQN that use polyak-averaging and delayed target network updates.

**Summary Of The Review:**

The paper is easy to read and provides a significant contribution to a very widely known problem. The solution can easily be integrated in existing applications and frameworks and is hence of broad interest to the community. While I am missing a few experimental benchmarks against approaches that address the same problem but differently, in my opinion the paper provides sufficient experimental evidence to judge the advantages and limitations of the proposed approach.

---

> ### Author Response · Authors · 2021-11-19
> **Response to Reviewer yew3**
>
> Thank you for your comments and suggestions.
>
> **I wondered how results in DDPG should look like – however, the results should also hold here.**
>
> For environments where Polyak averaging works well such as the one where DDPG is usually used, e.g. continuous control, the target network is constantly being (slowly) updated. Thus the target network is less a bottleneck than on the Atari benchmark where the target network is only updated every 40000 gradient steps. But indeed it is possible that FR improves DDPG in certain cases.
>
> **However, what I am missing a bit is an experimental benchmark against related approaches that have been proposed (e.g., Shao et al. (2020) and Kim et al. (2019) which have both been discussed as similar approaches in the paper). I don’t expect them to be much worse or much better either. The proposed functional regularization remains much more intuitive and tractable and also more computationally efficient (which at least for me would the killer-argument to used FR over other that trade performance in sample efficiency over ease of implementation).**
>
> GRAC (Shao et al.) has been developed for continuous control tasks, relies on CEM to choose the next actions, and thus makes the assumption that the actions are continuous. It allows GRAC to make “learning more robust to local noise in the Q function approximation”. Adapting GRAC to discrete actions would require new design choices. We have however included DeepMellow from Kim et al to the benchmark and updated Figure 2, Figure 3, and Figure 5.
>
> GRAC is indeed more expensive as they used CEM to learn the Q value. DeepMellow would not be more expensive, but the max entropy DeepMellow Asadi et al. [1] is indeed more expensive as they require access to Brent’s method in NumPy to tune the state-dependent temperature parameter.
>
> **Can you explain why in Sequest everything is different?**
>
> Seaquest is known to be particularly noisy, see https://raw.githubusercontent.com/deepmind/dqn_zoo/master/plot_atari_individual.svg In our case, one of the seeds got stuck with a sub-1000 return increasing variance and decreasing average performance.
>
> **Sec. 4.1.2: for the experiments in Fig. 2(a): do other DQN-variants that use other target network update periods lie in between those two graphs (10 and 100)? + Sec. 4.1.2: \tau=0.05 initially outperforms \tau=0.01 […]. Does it reach the plateau in the end (looks like it will but not sure)**
>
> We added full learning curves on 500k steps in the appendix. Note that these are new learning curves and were not used to select the hyperparameters in the four rooms environment.
>
>
> [1] https://arxiv.org/abs/1612.05628

---

> > ### Comment · Reviewer_yew3 · 2021-11-21
> > **Reply to author's response**
> >
> > I have read your response. Thank you for your answer and clarifications.

---

### Official Review · Reviewer_ugJ6 · 2021-11-02

**Correctness:** 4
**Technical Novelty And Significance:** 3
**Empirical Novelty And Significance:** Not applicable
**Recommendation:** 6
**Confidence:** 3

**Main Review:**

The approach is elegant and well-motivated.

Writing is clear.

The results in the four rooms task show good performance: compared to DQN and Polyak DQN, the proposed approach can recover the optimal value function quickly (given that the optimal value function in this task can be computed using tabular methods in this task). The proposed approach can match and sometimes get better than DQN / Polyak DQN on Atari tasks.


Comments/concerns:

More detailed analysis on how FR DQN performs on tasks with very sparse rewards would be interesting (perhaps also with a relatively large action space), given that this is one of the motivations. Will FR DQN actually perform better when there're sparse rewards?

How does FR DQN compare to "increasing the update frequency of target nerworks"?

More principled / automatic ways of adjusting kappa in Eq. (5) would be interesting. The kappa value is likely a very important term.

It’s not immediately clear to me why on some tasks, FR DQN results in a much larger standard deviation (seqquest).

It's not clear to me why FR DQN needs such a "target update period" shown in Table 1 in the appendix.

Minor: page 4 typo ("sectionSection"), page 6 typo ("move between room-").

**Summary Of The Paper:**

Given that in Q-learning, the Bellman error (which we optimize) involves the target Q values with ever-changing parameters, two major approaches have been proposed to address the instability: (1) have a copy of a slightly old version of Q (i.e., periodically update the parameters to estimate target Q); (2) use a moving average of parameters. Learning is slow in both cases. For example for (1), suppose the Markov chain has N states and we update the target network once every H steps, then it would take NH steps for a sparse reward (right-most reward of the chain) to propagate to the beginning. Similarly, weight regularization in DQN is ineffective too.

Therefore, the authors propose deep Q learning with functional regularization (FR DQN). See Eq. (5).

Essentially, they replace the target network Q’ by the current network Q (and adding stop-grad); and they add the regularization term at the end, i.e., (Q - Q’)^2, so that the up-to-date parameter can be used in the Bellman error.


**Summary Of The Review:**

The approach is well-motivated. The comparison to Polyak DQN is great. More experiments on complex tasks with relatively large action space, with sparse rewards will be great. A few minor concerns/questions in the main review.

---

> ### Author Response · Authors · 2021-11-19
> **Response to Reviewer ugJ6**
>
> Thank you for your comments and suggestions.
>
> **More detailed analysis on how FR DQN performs on tasks with very sparse rewards would be interesting (perhaps also with a relatively large action space), given that this is one of the motivations. Will FR DQN actually perform better when there're sparse rewards?**
>
> To test this hypothesis, we benchmarked FR DQN against the baseline on increasing Four Rooms environments. We show that FR dqn scales more gracefully than the baselines as we increase the environment size. Please refer to the general response or Figure 5 in the appendix for additional details.
>
> **How does FR DQN compare to "increasing the update frequency of target networks"?**
>
> FR DQN is not equivalent to increasing the update frequency of target networks. Increasing the frequency of target networks might increase the learning speed of DQN, but might also introduce instability and cause the objective to soft diverge. This can be explained by target networks providing both the learning signal and the regularization. FRDQN, on the other hand, decouples these two components by using the most up-to-date network to estimate the Q target, namely Q_\theta, and using a separate network to provide regularization, namely Q_{\bar{\theta}}.
>
> **More principled / automatic ways of adjusting kappa in Eq. (5) would be interesting. The kappa value is likely a very important term.**
>
> Indeed, automatically tuning kappa is an interesting direction for future work and could potentially improve performance. Currently, the additional parameter kappa will allow practitioners to tune regularization for their problem in a way that was impossible with target networks. In the same way, they are currently tuning batch size, learning rate, etc. to achieve the best performance on their particular problem.
>
> **It’s not immediately clear to me why on some tasks, FR DQN results in a much larger standard deviation (seqquest).**
>
> Seaquest is known to be particularly noisy, see https://raw.githubusercontent.com/deepmind/dqn_zoo/master/plot_atari_individual.svg
>
> **It's not clear to me why FR DQN needs such a "target update period" shown in Table 1 in the appendix.**
>
> This is indeed confusing, we added a row to the table to read prior update periods.

---

### Author Response · Authors · 2021-11-19
**General Response**

We want to thank all the reviewers for their thoughtful comments and questions. We also apologize for the delay in our answer as we had to run a lot of new experiments to answer all your questions and comments. For simplicity of reviewing, we added all the new figures to the appendix and we will add the important ones to the main text in the final version.

**Two reviewers ask about comparison with Kim et al. Deepmellow**

We have implemented DeepMellow in the dqn_zoo codebase. We performed careful hyper-parameter tuning on the 7 Atari games to find a temperature (we tested 25, 50, 75, 100) the exact same way we did it for our method FR-DQN, i.e., run 2 seeds for each hyper parameter and then report the result of the best hyper-parameter on 10 new seeds. Furthermore, we used the huber loss that is omitted in their paper, but could be seen in their implementation [1]. The learning rate, batch size, and reward clipping used in their implementation are the same as the one used in dqn_zoo. We list the differences with respect to their implementation that were not changed to have a fair comparison with DQN: the number of actions repeat (deepmellow uses 1 and dqn_zoo uses 4), observation dimensions (80x80 for deepmellow and 84x84 for DQN), and the exploration epsilon used for evaluation (0.01 for Deepmellow [2] and 0.05 for DQN). We note that DeepMellow is very sensitive to the temperature parameter and finding a temperature that works on every environment is very difficult e.g., in the DeepMellow paper they recommend a temperature of 30 for seaquest (20 and 40 achieve much worse performance) and 1000 for breakout. Note that we use temperature to refer to the inverse temperature of the Boltzman as done in Kim et al. We added these results in Figure 3 of the main text. Additionally, we ran experiments on Four Rooms with Deep Mellow. We added the curves to the appendix.

[1] https://github.com/seungchan-kim/DeepMellow/blob/4185a5d9c83385259902d53514a240a7cb3bd95e/Atari/deepmellow/agents/deep_q.py#L24
[2]
https://github.com/seungchan-kim/DeepMellow/blob/4185a5d9c83385259902d53514a240a7cb3bd95e/Atari/deepmellow/main.py#L179

**Combine FR with Rainbow.**

We understand that Rainbow is currently state of the art on Atari and it would be interesting to see if FR improves state-of-the-art methods. But Rainbow is built on top of distributional DQN which estimates the Q value as a weighted sum of atoms and then minimizes a cross-entropy loss to learn the Q value. This alternative cross-entropy loss is less popular than the squared Bellman error. Thus we believe that our paper extends the scientific knowledge of the community by focusing on the squared Bellman Error. In future work, it would be interesting to expand our regularization to arbitrarily loss functions and to Rainbow/distributional DQN.

**Sparse reward and the Pixel Four Room environment**

A reviewer showed interest in studying the effect of sparse reward on FR DQN compared with the baselines. We perform an additional ablation study for different sizes of the Pixel Four Room environment. The increase in size makes the reward sparser. We see that FR DQN scales more gracefully than the baselines as we increase the environment size. See Figure 5 in the Appendix for the regrets of each algorithm and additional learning curve on the initial four room environments.

**Convergence Proof**

Furthermore, we have demonstrated that value iteration with functional regularization induces a contraction and will converge to the true Q function for a given policy (Theorem 1 in appendix A.3), given some easily satisfied constraints on the learning rate and prior update period. We will add the theorem to the main text before publication.

---

### Decision · Program_Chairs · 2022-01-20

**Decision:**

Reject

**Comment:**

This paper presents a simple, reasonable, alternative to target networks.  Given the effectiveness of target networks, and the fact that they are still somewhat poorly understood, this is a good topic for consideration.

It is unfortunate that the paper did not have more depth, in terms of analysis and/or analytical experiments that expose the properties of the suggested approach, and the mechanism still seems heuristic (and inspired by the success of target networks, and similar) more than principled.  That said, the proposed mechanism does seem somewhat effective (even if performance differences are not very pronounced), and is clean and simple to implement.

This version of the paper is rejected because we believe the paper could be a lot better than it currently is.  If the proposed regularisation mechanism is really as good as the authors argue, then it should be possible (and hopefully even easy) to demonstrate this clearly in more settings (e.g., in more algorithms).  Alternatively or additionally, the authors could consider digging deeper into the understanding of the method.  For instance, the paper often argues that target networks slow down learning, but (naively?) one could argue the exact same point (in general) for regularisation: this will trade off stability for speed.  It could be that the proposed mechanism is indeed a better way to achieve this trade off, but this is currently argued heuristically and not really proven (either theoretically, or with sufficient empirical evidence)
(For what it is worth, I personally did not find Section 3.2 particularly enlightening, because it is known these TD algorithms are not actually gradient algorithms, and hence considering 'losses' and 'gradients' in this way does not convince me we are getting at an actual deeper understanding of the dynamics of these algorithms.)

I wholeheartedly encourage the authors to take the comments and suggestions to heart and use these to improve the paper (as they have already started to do during this reviewing cycle), because I believe that there could be quite a good paper on this topic.  I hope the authors can convince themselves and their readers more convincingly that this idea is an actual, lasting contribution to the literature.  Ultimately, if they can, this will make the paper more impactful.  So although I appreciate this decision will come as a disappointment, I hope the authors also see this as an opportunity to make a larger research impact.

In particular, I would encourage considering: 1) comparing to our current theoretical understanding of target networks (see, e.g., [1]); 2) considering the effect of multi-step updates (shown in, e.g., [2] and [3] to be quite effective); and 3) considering whether the proposed approach (or a variation thereof) could be understood as a more fundamental idea: could this update be derived from first principles?

[1] Shangtong Zhang, Hengshuai Yao, Shimon Whiteson (2021). Breaking the Deadly Triad with a Target Network.

[2] Hessel et al. (2017). Rainbow: Combining Improvements in Deep Reinforcement Learning.

[3] van Hasselt et al. (2018). Deep Reinforcement Learning and the Deadly Triad.